# A Hybrid Method for Calculating the Chemical Composition of Steel with the Required Hardness after Cooling from the Austenitizing Temperature

**DOI:** 10.3390/ma17010097

**Published:** 2023-12-24

**Authors:** Jacek Trzaska, Wojciech Sitek

**Affiliations:** 1Department of Engineering Materials and Biomaterials, Faculty of Mechanical Engineering, Silesian University of Technology, 44-100 Gliwice, Poland; 2Scientific and Didactic Laboratory of Nanotechnology and Material Technologies, Faculty of Mechanical Engineering, Silesian University of Technology, 44-100 Gliwice, Poland; wojciech.sitek@polsl.pl

**Keywords:** steel, artificial neural networks, genetic algorithm, optimization, materials by design, heat treatment, hardness

## Abstract

The article presents a hybrid method for calculating the chemical composition of steel with the required hardness after cooling from the austenitizing temperature. Artificial neural networks (ANNs) and genetic algorithms (GAs) were used to develop the model. Based on 550 diagrams of continuous cooling transformation (CCT) of structural steels available in the literature, a dataset of experimental data was created. Artificial neural networks were used to develop a hardness model describing the relationship between the chemical composition of the steel, the austenitizing temperature, and the hardness of the steel after cooling. A genetic algorithm was used to identify the chemical composition of the steel with the required hardness. The value of the objective function was calculated using the neural network model. The developed method for identifying the chemical composition was implemented in a computer application. Examples of calculations of mass concentrations of steel elements with the required hardness after cooling from the austenitizing temperature are presented. The model proposed in this study can be a valuable tool to support chemical composition design by reducing the number of experiments and minimizing research costs.

## 1. Introduction

Steel is one of the most important materials used in all sectors of the economy due to its good mechanical, physical, and functional properties. The selection of steel for structural components and machinery requires an analysis of the working conditions to ensure the required properties of these elements. In the case of structural and engineering steels, the required properties are achieved through the proper selection of the chemical composition of the steel and the appropriate choice of heat treatment, thermomechanical, or thermochemical processing conditions. Properly selected chemical composition should ensure the required steel properties and production costs. Knowledge of the qualitative and quantitative influence of alloying elements on steel structure and properties is essential for the rational selection of mass concentrations of these elements. The influence of an element should be considered in conjunction with other elements present in the steel, as they can significantly change the interaction with the structure and properties of the steel. The results of research on this subject have been published in numerous publications [1,2,3,4,5,6].

Continuous cooling transformation (CCT) diagrams provide significant information on the possibility of obtaining the required microstructure and hardness of the steel as a function of the cooling process from the austenitizing temperature. Dilatometric and metallographic methods are mainly used to develop these diagrams. CCT diagrams are often presented in a temperature–cooling-time format. They contain information on the start and end temperatures of the various phase transformations. Typically, the cooling rates of the samples and the hardness measured after cooling at these rates are plotted on the diagrams. The diagrams often also include the volume fractions of ferrite, pearlite, bainite, and martensite, which are also associated with specific cooling rates. CCT diagrams are used to determine the conditions for quenching, normalizing, and full annealing [7,8]. The parameters for phase transformation models are often calculated based on CCT diagrams. A CCT diagram is important for modeling and planning heat treatment and thermomechanical treatment of steel with known chemical composition and specific austenitizing conditions [9,10].

In recent years, the methods and tools for modeling and simulation of technological processes for manufacturing, processing, and shaping steel structures and properties have developed dynamically. Computer-aided modeling is used in scientific research and the industry. It is a relatively inexpensive and effective method to optimize, among others, the chemical composition and process conditions to achieve the required material properties [11,12,13,14]. The increasing availability of material databases and advances in machine learning methods are creating new opportunities for material design [15,16,17,18].

The growing interest in the application of artificial intelligence and computational intelligence in various fields of science and technology can also be observed in the field of materials engineering. Artificial neural networks are often used as a modeling method. Artificial neural networks are a useful tool for practical tasks. The use of artificial neural networks is especially justified when there are difficulties in creating mathematical models. Artificial neural networks make it possible to establish relationships between variables without defining a mathematical description of the analyzed problem. In the case of supervised learning, artificial neural networks learn to solve the problem based on examples [19]. The significant application potential of artificial neural networks in materials engineering has been presented by many authors [20,21,22,23,24].

A clearly visible trend in modeling, also in materials engineering, is the use of hybrid methods. Combining different methods in one model makes it possible to consider a broader problem space and achieve a synergistic effect by utilizing the advantages of each method. Artificial neural networks are often combined with other modeling methods, including mathematical modeling, computational intelligence, and artificial intelligence [25]. Combining artificial neural networks and genetic algorithms has become a favorable option to leverage the strengths of both methods. The combination of artificial neural networks and genetic algorithms enables the optimization of tasks. Artificial neural networks are used in this case to calculate the fitness values of individual chromosomes. The chromosomes represent the encoded values of the decision variables and form a set of possible solutions. Given the correct definition of the task conditions, the fitness value of the chromosomes corresponds to the value of the optimized objective function. This allows the identification of independent variables that meet the required criteria.

Examples of such solutions can be found, among others, in the works [26,27,28,29,30,31,32]. Reddy et al. [26] applied artificial neural networks and a genetic algorithm to optimize the chemical composition and heat treatment conditions of medium carbon steels with respect to the required mechanical properties: yield strength, ultimate tensile strength, elongation, area reduction, and impact strength. Dutta et al. [27] proposed a similar methodology to design the chemical composition of dual-phase steels. Pattanayak et al. [28] used artificial neural networks and a genetic algorithm to design the chemical composition and heat treatment conditions of microalloyed steels for pipe manufacturing. Sitek [29] presented a method to support the design of the chemical composition of high-speed steels with the required hardness and fracture toughness. Feng and Yang [30] proposed a method to optimize the thermomechanical processing conditions of austenitic stainless steel type 304 to increase resistance to intergranular corrosion. Razavi et al. [31] described a method for optimizing the heat treatment conditions of corrosion-resistant steels with an emphasis on maximizing hardness. Sinha et al. [32] focused their research on Ni-Ti shape memory alloys with the aim of improving shape recovery behavior while maintaining high mechanical properties. The authors of these publications emphasize that their proposed method reduces the number of experiments required to design new steel grades with the required properties.

The result of the initial work related to the modeling of the chemical composition of steel for the required hardness after cooling from the austenitizing temperature is presented in [33]. An artificial neural network was developed to calculate the hardness of steel based on its chemical composition and austenitizing temperature. In the next step, 6500 randomly generated chemical compositions of steel were evaluated for hardness at ten average cooling rates. Subsequently, the steel composition that best met the required criterion was selected.

The purpose of this article is to describe a method to calculate the chemical composition of steel for quenching and tempering with the required hardness after the product has been cooled from the austenitizing temperature. Artificial neural networks and a genetic algorithm were used in this method. The presented method consists of two stages. The first stage involves developing a hardness model. Artificial neural networks were used to develop the hardness model. Hardness is calculated based on the chemical composition of the steel and the austenitizing temperature. The hardness model developed using this method was presented in the work [34]. This article presents new modeling results. In the second stage, a genetic algorithm was used to identify the chemical composition of the steel with the required hardness. An application in which the model was implemented is described. Calculation examples are presented.

## 2. Hardness Model

The increasing popularity of machine learning is also reflected in numerous articles focusing on modeling the transformation of cooled austenite [35,36,37,38,39]. The authors presented models for calculating phase-transformation temperatures, as shown in CCT diagrams. However, the hardness of the steel is usually not considered in these models. A simpler and less costly method of determining the hardness of continuously cooled steel from the austenitizing temperature is the Jominy End-Quench test. Incorporating the results of the Jominy End-Quench test into models used for heat treatment simulations requires calculating the cooling rates at the analyzed points on the cooled object and assigning them corresponding distances from the quenched end of the sample [40]. Methods for calculating Jominy hardenability curves are presented, among others, in works [41,42,43,44,45,46]. However, the Jominy test has certain limitations. For example, the critical cooling rate of high-hardenability steel may be less than the minimal cooling rate of the Jominy specimen [47].

Modeling steel hardness using data from CCT diagrams allows us to link hardness to the cooling curve and phase transformations. On the other hand, the available data are limited, mostly in graphical form, and the values for time or cooling rate are challenging to digitize.

Equations developed through multiple regression analysis are also used to calculate steel hardness. A popular model used to calculate the hardness of continuously cooled steel from the austenitizing temperature is the Maynier model [48,49]. The model takes into account the influence of chemical composition, austenitizing temperature, and time. The Maynier equations can be used to calculate the characteristic cooling rates at 700 °C at which the following microstructural constituents are formed in the steel: 100%, 90% and 50% martensite, 90% and 50% bainite, and 90% and 100% ferrite and pearlite. The Maynier model was developed using data obtained from about 300 CCT diagrams. Equations for calculating the hardness of continuously cooled steel from the austenitizing temperature can also be found in this work [50]. The formulas were developed using data collected from approximately 500 CCT diagrams, employing multiple regression analysis and logistic regression. Based on the same dataset, a hardness model was developed using artificial neural networks, and the results were presented in the paper [34]. The same methodology was used for modeling in this study. Information from 50 additional CCT diagrams was added to the dataset, resulting in a new hardness model. In order to provide a comprehensive discussion of the methodology used to calculate the chemical composition of steel with the required hardness, some details of the hardness calculation method, as presented in [34], were reiterated.

### 2.1. Dataset for the Model

The development of a neural network model for steel hardness requires the preparation of a representative dataset of empirical data. The dataset was based on 550 CCT dia-grams published in the literature. The preparation of the dataset started with the determination of the variables representing the model. The selection of the independent variables should be supported by the knowledge of the modelled process. At the same time, the vectors containing examples of parameter calculation and model testing must contain values for all variables. Therefore, it was necessary to make simplifications regarding the number of independent variables. Information on austenitizing time and austenite grain size is not included in many CCT diagrams and was, therefore, not included in the model. During the cooling of steel from the austenitizing temperature, phase transformations that determine the microstructure and its hardness occur. The CCT diagrams contain information about the phase transformations that occur in the steel during cooling at a known rate. The information about the phase transformations was used in the hardness model.

It was established that the independent variables of the hardness model would be the mass concentration of the elements: C, Mn, Si, Cr, Ni, Mo, V, Cu, austenitizing temperature (T_A_), and cooling rate (CR). Additionally, four categorical independent variables describing the presence of ferrite, pearlite, bainite, and martensite in the steel structure were considered. The values of these variables were determined from the cooling curves presented in the CCT diagrams. The dependent variable of the model was the hardness of the steel obtained after cooling at a specified rate.

The values of the categorical variables that describe the steel structure were read from the CCT diagrams. However, the use of the model requires knowledge of these variables. Therefore, four classifiers were developed. The task of the classifiers was to answer the question of whether the steel with a certain chemical composition contains ferrite, pearlite, bainite, or martensite after cooling at a certain rate. The vectors of variable values for training and testing the classification neural networks contained the mass concentrations of elements, austenitizing temperature, cooling rate, and a categorical variable describing the occurrence of a phase transformation, such as ferritic (F), pearlitic (P), bainitic (B), and martensitic (M). The categorical variable could take one of two values: Yes or No.

To determine the applicability of the developed hardness model, an analysis of the range of values of the independent variables was performed. The examples prepared for the development and testing of the model covered the entire domain of approximated functions. Additional conditions were established that limit the use of the model.

The distributions of the values of the independent variables were assessed using descriptive statistics, scatter plots, and histograms for one and two variables. Descriptive statistics included minimum and maximum values, mean, standard deviation, median, skewness, and kurtosis. Attention was paid to outliers and collinearity of the independent variables. To test the correlation between variables, Pearson’s correlation coefficient was calculated for each pair of quantitative variables. For categorical variables, Spearman rank correlation was used. The results are presented in Figure 1.

Analyzing the data from Figure 1, a moderate positive correlation can be observed between the variable describing the occurrence of martensite in steel and the cooling rate. A similar relationship holds for ferrite and pearlite. There is a moderate negative correlation between the variables that describe the proportion of ferrite and pearlite and the cooling rate. A similar relationship exists between the variables describing the occurrence of ferrite and pearlite in steel and the variable describing the occurrence of martensite. The negative value of the coefficient confirms that, in most cases, these components do not occur simultaneously in the steel. There is a small negative correlation between the concentration of chromium and manganese in steel. The highest value of the correlation coefficient for steel hardness is observed for the categorical variables (F, P, M), cooling rate, and carbon concentration. This confirms the knowledge about the influence of the model variables on the hardness of the steel cooled from the austenitizing temperature.

For the cooling rate, a transformation was applied that involved the calculation of the fourth root. This provided a uniform distribution of variable values over the entire range.

The values of the input variables and the output variables were scaled in the range of 0 to 1 using the min–max function.

The hardness model can be applied within the range of mass concentrations of the elements listed in Table 1. Based on the statistical analysis of the data, additional conditions limiting the use of the model were defined. The additional constraints are presented in Table 2.

A verification set consisting of 25 CCT diagrams was created. The data from this set were not used to calculate the model parameters. They were used only for numerical verification of the developed relationships.

The dataset prepared for model development was divided into a training dataset, a validation dataset, and a test dataset. The training dataset was used to determine the weights of connections between neurons during training. The validation dataset was used to evaluate the neural network during training. The test dataset was used to evaluate the quality of the neural network after training. Several vectors of variables were obtained from one CCT diagram, ranging from a few to several. Random division into training, validation, and testing datasets resulted in examples derived from a CCT diagram being assigned to different sets. Using the data from the verification set, it was possible to compare the overall hardness change curve for steel with a given chemical composition. The training, validation, test, and verification datasets contained 1763, 550, 550, and 300 patterns, respectively.

### 2.2. Methods and Results

According to the assumptions adopted, the hardness neural model consisted of five neural networks, including four classifiers. More information on neural classifiers is presented in [51]. The process flow of this method is shown in Figure 2. One of the independent variables of the model is the austenitizing temperature. It was assumed that the austenitizing temperature would take the value of A_c3_ + 50 °C. The value of the end temperature of the A_c3_ transformation during heating was calculated based on the chemical composition using a neural network. The neural network model for calculating the A_c3_ temperature is presented in [51].

Artificial neural networks were designed and tested using the STATISTICA Neural Networks 4.0 F software.

To assess the quality of neural networks in regression tasks, the following statistics were used: mean absolute error, standard deviation of error, Pearson’s correlation coefficient, and standard deviation. The ratio of the standard deviation of the prediction error to the standard deviation of the dependent variable allows a comparison of the error values made by the neural network with the range of values of the dependent variable. A smaller prediction error and a larger range of the dependent variable result in smaller values of the standard deviation ratio, reaching zero for a perfect prediction. These statistics were calculated for the training, validation, test, and verification datasets.

In the case of classifiers, the two classes (presence or absence of a transformation) were represented in the dataset by a similar number of examples. Evaluation of the classifiers involved the use of the accuracy coefficient and Receiver Operating Characteristic (ROC) curve. The accuracy coefficient was calculated as the ratio of the number of correctly classified cases to the total number of cases in the dataset. The ROC curve allows one to evaluate the performance of a binary classifier for all possible thresholds that determine the class boundary. For random classification, the area under the ROC curve (AUC) is 0.5, while for a perfect classifier, it reaches a value of 1.

In the initial modeling phase, several types of feedforward neural networks were used: linear networks, multilayer perceptrons (MLP), radial basis functions (RBF), generalized regression neural networks (GRNNs) for regression tasks, and probabilistic neural networks (PNNs) for classification tasks only. After preliminary calculations were performed and the results obtained were analyzed, it was decided that only MLP networks with a single hidden layer would be considered in the next stage of work. The focus was on determining the optimal number of neurons in the hidden layer combined with training the neural network.

Artificial neural networks were trained using the following methods: backpropagation (BP), quick propagation (QP), conjugate gradients (CG), Levenberg–Marquardt (LM), quasi-Newton (QN), and delta-bar-delta (DD). During the neural networks training, the root mean square error (RMSE) value was analyzed. The change in RMSE value was observed in successive training epochs for the training and validation datasets. The training of the network was stopped in the epoch in which the error of the validation dataset started to increase. In a typical artificial neural network training process, after a certain number of training epochs, despite a decrease in the error value for the training set, the error for the validation set starts to increase. In such cases, continuing the training leads to overfitting the model to the training data. The detrimental effect of overfitting occurs relatively frequently in neural networks. Overfitting is favored by an increase in the number of hidden layers, an increase in the number of neurons in the hidden layer(s), and an excessive increase in the number of training epochs.

After completion of the neural network, the aforementioned statistics used for evaluating the neural network were calculated. These calculations were performed for the training, validation, test, and verification sets. A series of experiments were performed varying the number of neurons in the hidden layer as well as the method and/or training parameters.

The selection of the neural network that best describes the relationship between independent variables and steel hardness was based on the values of the statistics mentioned above (regression tasks). Attention was paid to ensure that the values of the respective statistics for the four sets were similar. In the case of neural networks with similar statistical values, the network with fewer neurons in the hidden layer was chosen. The best fit was obtained for the MLP network with a structure of 14-8-1. The neural network was trained using the Levenberg–Marquardt method for 172 training epochs. The sum of squares was used as the error function, and the activation functions in the input, hidden, and output layers were linear, logistic, and linear, respectively. The statistical values used to evaluate the neural network are summarized in Table 3.

The parameters characterizing the neural networks developed to identify the structural components present in steel after cooling are presented in Table 4. The metric values used to evaluate the classifiers were collected in Table 5. The input variables of all classifiers were mass concentrations of elements: C, Mn, Si, Cr, Ni, Mo, V, Cu, and T_A_ and CR.

The significance of the independent variables was evaluated by the ratio between the estimated error of a neural network without the influence of the analyzed variable and the error of the neural network considering the influence of all input variables. When estimating the error of the neural network without the influence of the independent variable, the mean value of this variable is assumed for all patterns. An independent variable was considered significant if the calculated ratios for the training set and the validation set were greater than 1. The error ratios calculated for the training and validation sets are shown in Figure 3.

The obtained results confirm that the categorical variable describing the martensite content in the steel structure, the mass concentration of carbon, and the cooling rate has the greatest influence on the accuracy of the prediction. This supports the consideration of including categorical variables describing phase transformations during steel cooling in the hardness model. Categorical variables significantly reduce prediction error and compensate for the additional cost of introducing classifiers into the model. Note that the operation of classifiers is subject to error. However, their inclusion in the model has noticeably reduced the computational error.

The results presented demonstrate that the hardness model can be applied to calculate the hardness of steel after the temperature is cooled from austenitizing. When using the model, it is important to keep in mind the potential error that can occur during the calculations.

## 3. Calculating the Chemical Composition of Steel

### 3.1. ANN–GA Hybrid Model

The calculation of the chemical composition of steel with the required properties can be considered as the search for values of independent variables for which the objective function approaches the expected value. The values of the independent variables must be in the set of allowed solutions. The value of the objective function is calculated using a neural network model. In such cases, the space of possible solutions must be limited by the range of mass concentrations of the elements to which the neural network model can be applied. The constraints on the solution space are presented in Table 1 and Table 2.

Steel, with a known chemical composition, that is austenitized under the same conditions exhibits a unique hardness change curve as a function of the cooling rate. However, there are many steels for which the hardness curve takes on a similar shape. Therefore, there may be multiple solutions to calculate the chemical composition of steel with a required hardness change curve. The optimization method used to find a solution must be an efficient global method capable of obtaining multiple suboptimal solutions. Genetic algorithms are stochastic algorithms capable of solving the suboptimal solutions that are selected to optimize the parameters.

The essence of genetic algorithms, similar to other evolutionary methods, lies in the search for a solution within a limited space. This search is inspired by the mechanisms of natural selection and evolution. In the space defined by the constraints of the optimization problem, there exists a population of individuals encoded as potential solutions. In a classical genetic algorithm, the individuals are represented by chromosomes encoded as binary strings. The quality of these solutions is evaluated on the basis of the fitness function. Individuals with higher fitness, which better meet the search criteria, have a greater chance of survival and produce a new generation. Individuals exchange information through crossover and mutation operators. Through operations inspired by natural evolution, they create progressively better solutions in each iteration. The space of potential solutions is explored in parallel. The genetic algorithm follows an evolutionary rule in which the individuals with the highest fitness values have the highest probability of survival [52,53].

The objective of the optimization presented in this study was to identify the chemical composition of the steel with the required hardness after the steel is cooled from the austenitizing temperature. The required hardness is the hardness that should be achieved at five specified cooling rates of the element from the austenitizing temperature. The objective function was defined as a measure of the fitting error to the required hardness of the steel. The objective function is described by Equation (1). The calculations sought to minimize the value of the objective function. The required hardness was assumed to be determined for a maximum of five cooling rates. Each required hardness was assigned a weighting coefficient, which describes the significance of the hardness obtained at a specific cooling rate and can take a value from 0 to 1. The fitting error was calculated as the absolute difference between the calculated and required values. The scaling was carried out using the Min–Max function in the range from 0 to 1. The minimum and maximum hardness values were determined from the data used for the training of the neural networks.
(1)$$f_{HV}\left(x\right)=\sum_{i=1}^{k}w_{HVi}\cdot \left|\frac{\left({HVc}_{i}-{HV}_{min}\right)-\left({HVr}_{i}-{HV}_{min}\right)}{{HV}_{max}-{HV}_{min}}\right|$$
where:i = 1, 2,..., k;k = 1, 2,..., 5;w_HVi_—weighting coefficient for the hardness at the i-th cooling rate;HVci, HVri—the calculated or required hardness for the i-th cooling rate;HV_min_, HVmax, the minimum and maximum hardness determined based on empirical data analysis;x—vector of independent variables.

The use of genetic algorithms for the identification of the chemical composition of steel and artificial neural networks to calculate the fitness function requires the development of a computer program. The program was written in the C++ language. The calculation algorithm performed by the computer program is presented in Figure 4.

The program implemented a classical genetic algorithm with binary chromosome encoding. It was assumed that the length of the binary string for each variable is 10 bits so that the value range of the variables can be divided into 1024 intervals. This value can be reduced if necessary. The roulette wheel selection method, single-point crossover, and mutation operators were applied. The fitness function was defined according to Equation (1). Neural networks were used to calculate the value of the fitness function. Artificial neural networks trained to calculate the hardness of steel were defined as functions of the code. This approach allows easy modification of the program in case neural networks with lower computational errors are developed. In the calculation of genetic algorithms, premature convergence can be observed, which is caused by the dominance of the fittest individuals. This phenomenon often occurs in proportional selection methods, such as roulette wheel selection. After several generations, the population may consist only of copies of the best chromosome. In the final phase of the algorithm, there is often a small difference between the average fitness value of the population and the fitness value of the best individuals. This situation reduces the competition among individuals and can lead to a genetic drift effect. To avoid premature convergence of the algorithm, power-law scaling is utilized to scale the fitness value.

After defining the required hardness as a function of the cooling rate, it is possible to calculate the mass concentration of all elements or only selected elements. Algorithm parameters that can be adjusted during the calculations include the number of generations, the size of the population, the probability of crossover, and the probability of mutation. An elitist strategy was applied in the reproduction procedure, which involves including the best individuals unchanged in the next generation. The aim of the elitist strategy is to preserve the best chromosomes in successive generations. The number of unchanged individuals is one of the program options and can be set to 0. The program interface is shown in Figure 5.

### 3.2. Examples of Applications of the ANN–GA Model

This paper presents three examples of the application of the developed method for calculating the chemical composition of steel with the required hardness after cooling from the austenitizing temperature.

In the first example, the calculations were limited to the carbon concentration. In order to verify the calculation results, the required hardness values were determined using the CCT diagram of the 41Cr4 steel [54]. The range of concentrations for other elements was limited to the chemical composition of the 41Cr4 steel and did not exceed 0.05%. Table 6 shows the hardness of 41Cr4 steel obtained from the CCT diagram, the required hardness, the calculated hardness, and the sum of the hardness errors. The error for each of the five cooling rates was calculated as the absolute difference between the required hardness value and the calculated hardness value. The table shows the sum of the absolute errors for the five cooling rates. Table 7 presents the chemical composition of the 41Cr4 steel, the calculated chemical composition, and the austenitizing temperature. Figure 6 shows the hardness curves as a function of cooling time based on the values in Table 7. The calculations were performed multiple times by changing the parameters of the genetic algorithm. The results presented were obtained with the following parameters: 1000 generations, population size of 200, crossover probability of 0.8, and mutation probability of 0.1.

The calculated carbon mass concentration in this example is 0.42. This concentration is 0.02% higher than the carbon concentration in the reference steel 41Cr4. There are also slight differences in the concentrations of other elements, such as manganese (0.02% difference). The hardness error obtained is 57 HV. The largest difference occurs at the highest cooling rate, with a difference of 10 HV. Similar calculations were performed for other steels from the validation set, calculating the concentrations of various elements. The results confirm a high level of agreement with the experimentally obtained results.

In the second example, the concentrations of two elements, Cr and Mn, were calculated. In this case, a comparison with the experimental results is only possible after melting and testing. The reference steel chosen in this example is 37Cr4 [55]. The range of concentrations for other elements was limited to the chemical composition of the 37Cr4 steel and did not exceed 0.05%. Table 8 shows the hardness of 37Cr4 steel obtained from the CCT diagram, the required hardness, the calculated hardness (various solutions), and the sum of the hardness errors. Different solutions were obtained after each successive run of the program parameters, with optional changes to the genetic algorithm. Five solutions are presented. Figure 7 shows the hardness curves as a function of cooling time based on the values in Table 8. Table 9 presents the chemical composition of the 37Cr4 steel, the calculated chemical compositions, and the austenitizing temperature.

The calculated results are close to the expected values. There is a relationship between the mass concentrations of Mn and Cr. An increase in the concentration of one element leads to a decrease in the concentration of the other. The sum of the element concentrations varies between 1.44 and 1.69. The sum of hardness error ranges from 30 to 62 HV.

In the third example, the concentrations of all elements were calculated. The required hardness was defined using the CCT diagram of 25CrMo4 steel [55]. Table 10 shows the hardness of 25CrMo4 steel obtained from the CCT diagram, the required hardness, the calculated hardness (various solutions), and sum of the hardness errors. Similar to the second example, the program was run multiple times with optional changes to the parameters of the genetic algorithm. Five solutions are presented. Figure 8 shows the hardness curves as a function of the cooling time based on the values in Table 10. Table 11 presents the chemical composition of the 25CrMo4 steel, the calculated chemical compositions, and the austenitizing temperature.

The chemical compositions of the steel represent only a subset of the possible solutions. The mass concentrations of carbon, chromium, nickel, and manganese undergo changes primarily. This is due to the influence of these elements on the hardenability and the distribution of values in the dataset. In each case, the calculated hardness is close to the required hardness. In this case, the sum of hardness error ranges from 9 to 32 HV. Numerical verification is not possible. Steels with similar chemical compositions are not present in the dataset used to train and test the artificial neural networks. The results obtained can only be verified by experiments.

## 4. Conclusions and Future Work

This paper presents a method for calculating the chemical composition of steel with the required hardness values after continuous cooling from the austenitizing temperature. A hybrid system consisting of artificial neural networks and genetic algorithms was used to identify the mass concentrations of elements. To perform the calculations, a computer program was developed that included a classical genetic algorithm to optimize the chemical composition of steel and a neural network model to calculate the values of the fitness function. The program has a graphical user interface that allows the user to define criteria for the solution search, constrain variable ranges, change the parameters of the genetic algorithm, and save the results obtained. The compatibility of the calculations with the experimental results was demonstrated in an example in which the mass concentration of a selected element was calculated.

In this version of the program, the user defines the hardness as a single value for each cooling rate. The fitting error is calculated as the absolute difference between the expected hardness and the calculated hardness. Consequently, the calculated hardness can be greater or less than the expected value. A planned extension of the program is to allow the user to define a hardness range by specifying minimum and maximum values. In such cases, the calculated chemical composition of the steel should ensure that the hardness falls within that range.

The final stage of modeling should include the experimental verification of the developed models. For the presented method, it is necessary to perform steel melts with the calculated chemical composition. In order to obtain reliable results, the sample should be sufficiently large. At this point, it should be noted that the article is a report on the completed phase of the planned work.

Work is currently underway on a multi-criteria optimization of the chemical composition, taking into account the phase transformation temperatures. The results obtained, which are limited to two cooling rates and the initial phase transformation temperature, are currently satisfactory. In future work, the number of cooling rates will be increased, and the end temperatures of phase transformation will be introduced. The expected result will be a method capable of calculating the chemical composition of the steel to match the required CCT diagram. Following this phase, an experimental verification will be carried out, including steel melting and dilatometric tests.

Designing the chemical composition of low-alloy steels with the desired properties is a complex process. The hardness of the steel after cooling from the austenitizing temperature is one of several essential criteria. The application of the proposed model can reduce the costs and the number of required experiments. The results obtained can be used for further analyses.

## Figures and Tables

**Figure 1 materials-17-00097-f001:**
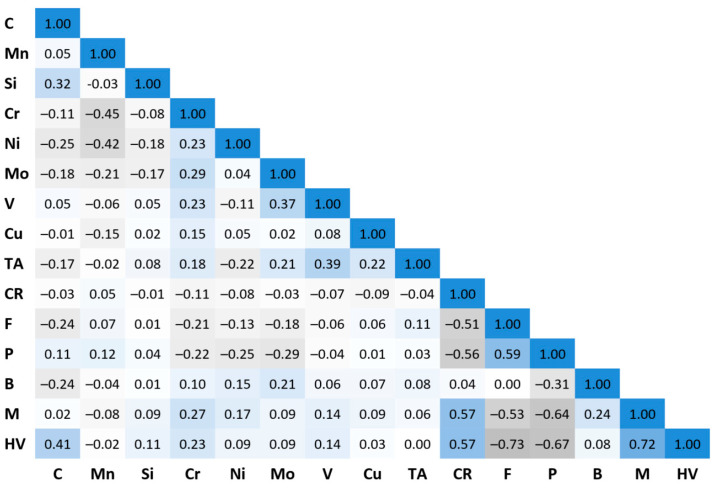
Correlation plot of all variables in the dataset.

**Figure 2 materials-17-00097-f002:**
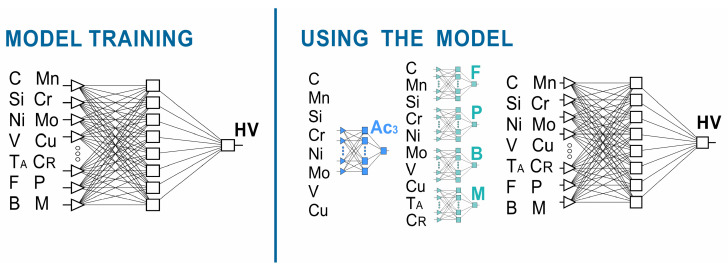
Flow chart of the methodology adopted for hardness calculation.

**Figure 3 materials-17-00097-f003:**
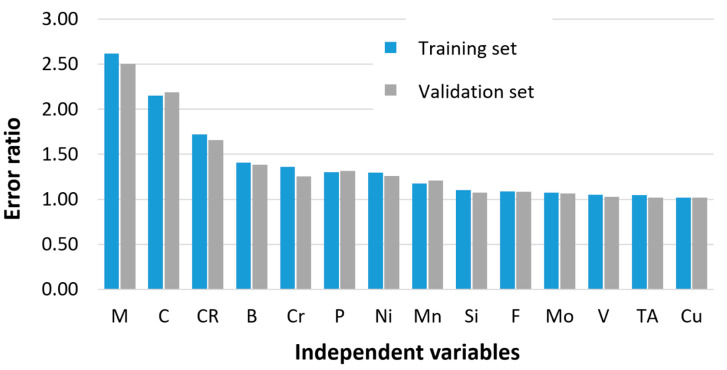
Impact of independent variables on steel hardness evaluated on the error ratio.

**Figure 4 materials-17-00097-f004:**
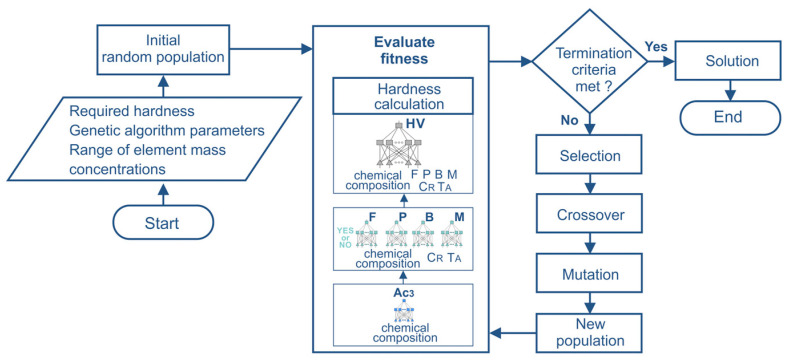
Flowchart of the optimization scheme based on the hybrid ANN–GA algorithm.

**Figure 5 materials-17-00097-f005:**
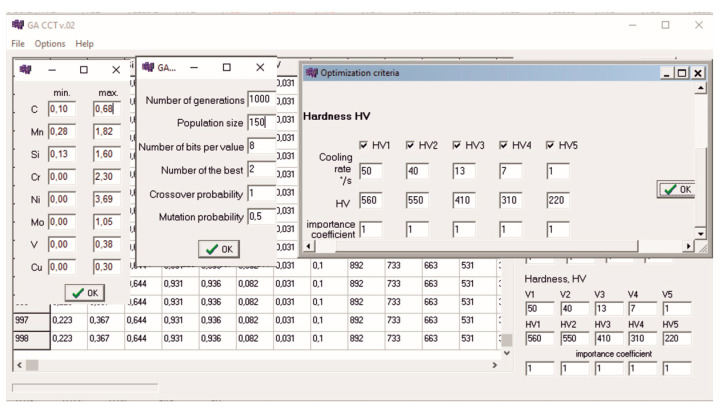
Computer app calculating the chemical composition of steel of assumed hardness.

**Figure 6 materials-17-00097-f006:**
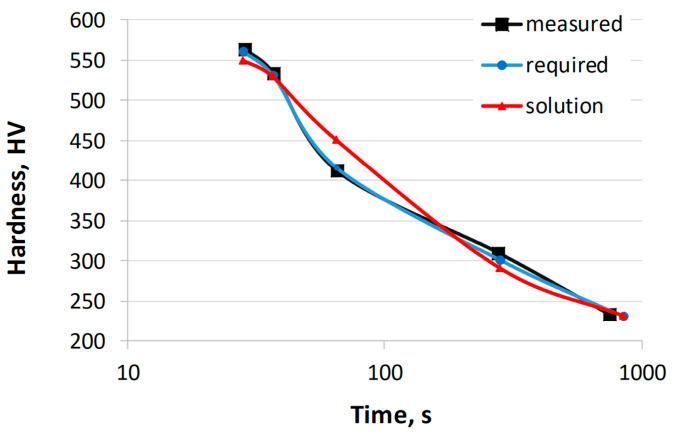
Comparison of hardness curves: experimental, required, and calculated (solution from Table 6 and Table 7).

**Figure 7 materials-17-00097-f007:**
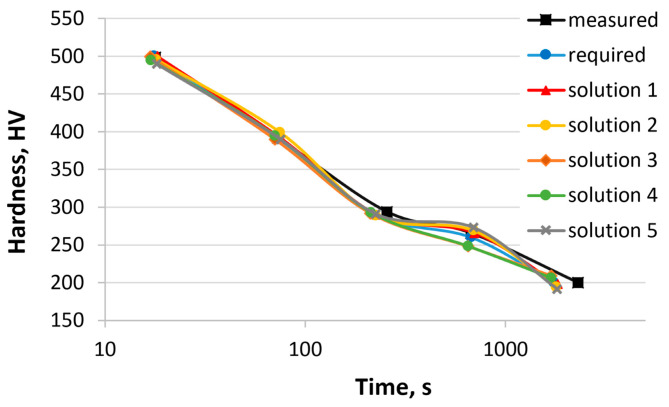
Comparison of hardness curves: experimental, required, and calculated (solutions 1–5, Table 8 and Table 9).

**Figure 8 materials-17-00097-f008:**
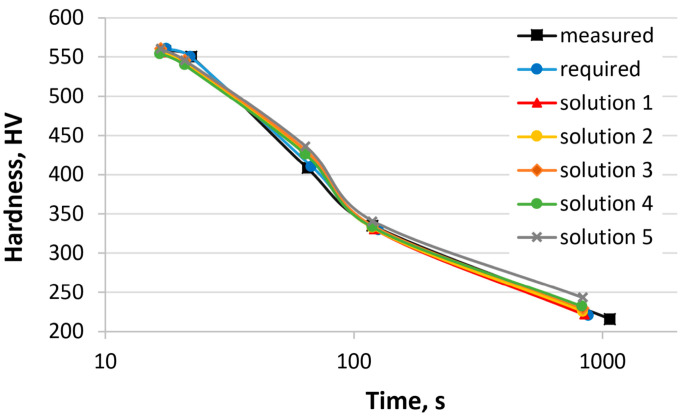
Comparison of hardness curves: experimental, required, and calculated (solutions 1–5, Table 10 and Table 11).

**Table 1 materials-17-00097-t001:** Minimum, maximum, mean, and standard deviation values of the input variables.

Variables	Minimum	Maximum	Mean	Std. Dev
C (wt%)	0.10	0.68	0.32	0.14
Mn (wt%)	0.25	1.80	0.79	0.33
Si (wt%)	0.13	1.60	0.33	0.28
Cr (wt%)	0	2.30	0.72	0.56
Ni (wt%)	0	3.60	0.74	1.00
Mo (wt%)	0	1.00	0.16	0.20
V (wt%)	0	0.38	0.02	0.06
Cu (wt%)	0	0.30	0.04	0.08
T_A_ (°C)	770	1050	878	57

**Table 2 materials-17-00097-t002:** Additional conditions limiting the scope of the model application.

	Mn + Cr	Mn + Cr + Ni	Cr + Ni	Mn + Ni
Maximum (wt%)	3.6	5.6	5.3	4.5

**Table 3 materials-17-00097-t003:** The statistical values used to evaluate the hardness model.

Dataset	Mean Absolute Error, HV	Standard Deviation of the Error, HV	Ratio of Standard Deviations	Pearson Correlation Coefficient
Training	30.9	44.3	0.27	0.96
Validating	33.6	46.4	0.28	0.96
Testing	33.7	50.1	0.30	0.95
Verifying	32.7	39.0	0.29	0.95

**Table 4 materials-17-00097-t004:** Structure and training parameters of neural classifiers.

	Transformation			
	Ferritic	Pearlitic	Bainitic	Martensitic
ANN structure	MLP 10-8-1	MLP 10-8-1	MLP 10-10-1	MLP 10-6-1
Training/No of epoch	BP/50, CG/330	BP/50, CG/119	BP/50, CG/188	CG100

**Table 5 materials-17-00097-t005:** Accuracy and AUC values used to evaluate neural classifiers.

Metric	Dataset	Transformation			
		Ferritic	Pearlitic	Bainitic	Martensitic
Accuracy	Training	0.92	0.92	0.86	0.89
	Validating	0.91	0.92	0.86	0.86
	Testing	0.89	0.91	0.84	0.86
AUC	Training	0.97	0.97	0.93	0.95
	Validating	0.96	0.97	0.92	0.94
	Testing	0.96	0.97	0.91	0.93

**Table 6 materials-17-00097-t006:** The required and calculated hardness of the steel after cooling at selected rates (Example 1).

	Cooling Rate, °/s					Sum of the Errors, HV
	30	23	13	3	1	
41Cr4	563	534	412	310	233	-
Target	560	530	415	300	230	-
Solution	550	530	451	291	232	57

**Table 7 materials-17-00097-t007:** Chemical composition of steel calculated for the required hardness (Example 1).

Variables	41Cr4	Solution
C (wt%)	0.40	0.42
Mn (wt%)	0.60	0.58
Si (wt%)	0.33	0.33
Cr (wt%)	0.93	0.94
Ni (wt%)	0.05	0.01
Mo (wt%)	0.00	0.04
V (wt%)	0.00	0.04
Cu (wt%)	0.09	0.04
T_A_ (°C)	850	841

**Table 8 materials-17-00097-t008:** The required and calculated hardness of the steel after cooling at selected rates (Example 2).

	Cooling Rate, °/s					Sum of the Errors, HV
	50	40	13	7	1	
37Cr4	558	550	408	335	216	-
Target	560	550	410	330	220	-
Solution 1	556	542	429	331	222	34
Solution 2	557	543	428	332	226	36
Solution 3	560	546	431	335	230	40
Solution 4	554	540	426	333	232	47
Solution 5	560	546	436	341	243	64

**Table 9 materials-17-00097-t009:** Chemical compositions of steel calculated for the required hardness (Example 2).

Variables	37Cr4	Solution 1	Solution 2	Solution 3	Solution 4	Solution 5
C (wt%)	0.38	0.40	0.40	0.40	0.40	0.40
Mn (wt%)	0.74	0.50	0.69	0.87	0.93	1.30
Si (wt%)	0.26	0.29	0.25	0.25	0.25	0.25
Cr (wt%)	0.90	0.94	0.87	0.72	0.61	0.39
Ni (wt%)	0.26	0.26	0.29	0.28	0.29	0.26
Mo (wt%)	0.04	0.03	0.01	0.00	0.01	0.03
V (wt%)	0.00	0.05	0.03	0.05	0.04	0.05
Cu (wt%)	0.07	0.04	0.05	0.05	0.05	0.04
T_A_ (°C)	880	843	835	832	831	832

**Table 10 materials-17-00097-t010:** The required and calculated hardness of the steel after cooling at selected rates (Example 3).

	Cooling Rate, °/s					Sum of the Errors, HV
	50	12	4	1.3	0.5	
25CrMo4	498	392	294	266	200	-
Target	500	390	290	260	200	-
Solution 1	500	391	290	266	198	9
Solution 2	496	399	289	269	194	29
Solution 3	499	390	292	248	209	24
Solution 4	495	395	293	248	206	31
Solution 5	490	390	291	273	192	32

**Table 11 materials-17-00097-t011:** Chemical compositions of steel calculated for the required hardness (Example 3).

Variables	25CrMo4	Solution 1	Solution 2	Solution 3	Solution 4	Solution 5
C (wt%)	0.22	0.25	0.25	0.32	0.30	0.21
Mn (wt%)	0.64	1.20	1.00	0.59	0.50	1.41
Si (wt%)	0.25	0.35	0.45	0.31	0.37	0.31
Cr (wt%)	0.97	0.32	0.63	0.74	0.87	0.65
Ni (wt%)	0.33	0.30	0.48	0.78	1.01	0.20
Mo (wt%)	0.23	0.11	0.02	0.10	0.05	0.00
V (wt%)	0.01	0.35	0.28	0.02	0.00	0.35
Cu (wt%)	0.16	0.16	0.07	0.05	0.04	0.15
T_A_ (°C)	875	910	900	846	849	903

## Data Availability

The dataset used to support the findings of this study is available from the corresponding author upon request.

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
