# Peer review of "A Hybrid Method for Calculating the Chemical Composition of Steel with the Required Hardness after Cooling from the Austenitizing Temperature"

_materials, 2023, doi:10.3390/ma17010097_

Round 1

Reviewer 1 Report

Comments and Suggestions for Authors

1.      The purpose of this article is to describe a method to calculate the chemical composition of steel with the required hardness after the product has been cooled from the austenitizing temperature. Comments: Actually, the hardness depends on the cooling rate for the same chemical composition of steel.

2.      The statistical values used to evaluate the neural network are summarized in Table 3. Comments: the hardness value is suggested to be added.

3.      Table 6. check if the hardness of the steel after cooling at rate of 3°/s is only 30HV?

4.      Table 7, 9, 11. Chemical composition of steel calculated for the required hardness. Comments: need clearly show what the required hardness is.

5.      Line 86: ….are currently satisfactory [18].  Comments: reference should not be cited in the section of “Conclusions and future work”.

6.      English needs to be improved, there exist some grammar errors. Eg. 1) Line 86: Examples of such solutions can be found, among others, in the works [26–32]. Reddy et al. [26] applied artificial neural networks and…. 2) Line 131: Methods for calculating Jominy hardenability curves are presented, among others, in works [41–46] etc… 

Comments on the Quality of English Language

English needs to be improved, there exist some grammar errors. Eg. 1) Line 86: Examples of such solutions can be found, among others, in the works [26–32]. Reddy et al. [26] applied artificial neural networks and…. 2) Line 131: Methods for calculating Jominy hardenability curves are presented, among others, in works [41–46] etc… 

Author Response

Dear Reviewer,

Thank you for your comments concerning our manuscript entitled “A Hybrid Method for Calculating the Chemical Composition of Steel with the Required Hardness After Cooling from the Austenitizing Temperature”. We have studied comments carefully and prepared explanations, which we hope meet with approval. We appreciate the time and effort that you dedicated to providing feedback on our manuscript and are grateful for the insightful comments.

Best regards
Authors

Comments and Suggestions for Authors

  1. The purpose of this article is to describe a method to calculate the chemical composition of steel with the required hardness after the product has been cooled from the austenitizing temperature. Comments: Actually, the hardness depends on the cooling rate for the same chemical composition of steel.

Thank you for pointing this out. Of course we agree with the comment that the hardness varies with the cooling rate for the same chemical composition. These changes are illustrated by the hardness change curve as a function of cooling rate. The goal of the study is to calculate the chemical composition of a steel so that the hardness curve matches the required (expected) curve as closely as possible. It is worth noting that the cooling rate varies over the cross-section of the cooled element from the austenitizing temperature onwards

  1. The statistical values used to evaluate the neural network are summarized in Table 3. Comments: the hardness value is suggested to be added.

Thank you for this suggestion. All values in Table 3 relate to hardness.

  1. Table 6. check if the hardness of the steel after cooling at rate of 3°/s is only 30HV?

Thank you for pointing this out. It should be 300 HV. The mistake has been corrected.

  1. Table 7, 9, 11. Chemical composition of steel calculated for the required hardness. Comments: need clearly show what the required hardness is.

Thank you for this suggestion. The required hardness is the hardness value we want to achieve after cooling at a certain rate.

One sentence has been added to the manuscript:

The required hardness is the hardness that should be achieved at five specified cooling rates of the element from the austenitizing temperature.

  1. Line 86: ….are currently satisfactory [18]. Comments: reference should not be cited in the section of “Conclusions and future work”.

Thank you for this suggestion. The reference has been removed.

  1. English needs to be improved, there exist some grammar errors. Eg. 1) Line 86: Examples of such solutions can be found, among others, in the works [26–32]. Reddy et al. [26] applied artificial neural networks and…. 2) Line 131: Methods for calculating Jominy hardenability curves are presented, among others, in works [41–46] etc…

The sentences given by the reviewer as examples of incorrect grammatical form are correct in English. Since the reviewer did not give the correct form in his comment, the authors assume by default that the comment concerns the phrase "among others". The phrase used can function in three different forms - at the beginning, in the middle and at the end of the sentence. It depends on the context and the subject in the sentence. If you are referring to people, it is correct to use the phrase "among others" in the middle of the sentence when referring to a person (quoted literature).

Source:

https://grammarhow.com/among-others-meaning-usage/?utm_content=cmp-true

Reviewer 2 Report

Comments and Suggestions for Authors

This paper presents a method for calculating the chemical composition of steel with the required hardness values after continuous cooling from the austenitizing temperature. For that purpose, computer program based on artificial neural networks and genetic algorithms was developed. Artificial neural networks and genetic algorithms model is based on 550 diagrams of continuous cooling transformation (CCT) of structural steels available in the literature. Paper shows examples of calculations of chemical composition of steel with the required hardness after cooling from the austenitizing temperature. The model proposed by this study will be a valuable tool to support chemical composition design of steel.

Author Response

Dear Reviewer,

Thank you for your comments on our manuscript entitled “A Hybrid Method for Calculating the Chemical Composition of Steel with the Required Hardness After Cooling from the Austenitizing Temperature”. We appreciate the time and effort you took to provide feedback on our manuscript and thank you for the insightful comments. All corrections have been made according to your suggestions, which are listed in the file peer-review-33911556.v2.pdf.

Best regards

Authors

Reviewer 3 Report

Comments and Suggestions for Authors

The modelling needs to be explained into more detail, the theoretical results must be correlated better with the experimental values, there is no information on the errors/deviations on the actual parameters studied (chemical composition, hardness). How is the error ratio calculated? (information can be provided in a supplementary file).
Compare your model to other available ones from the reference literature

Author Response

Dear Reviewer,

Thank you for your comments on our manuscript entitled “A Hybrid Method for Calculating the Chemical Composition of Steel with the Required Hardness After Cooling from the Austenitizing Temperature”. We thank you for the time and effort you have taken to provide feedback on our manuscript.

Best regards
Authors

Comments and Suggestions for Authors

The modelling needs to be explained into more detail, the theoretical results must be correlated better with the experimental values, there is no information on the errors/deviations on the actual parameters studied (chemical composition, hardness). How is the error ratio calculated? (information can be provided in a supplementary file).

Compare your model to other available ones from the reference literature

Thank you for your comments and suggestions. We hope that our explanations will change your opinion of our publication, similar to the three other reviews, and lead to a positive outcome.

To estimate the parameters of the hardness model, we use data obtained from experiments. These include: the chemical composition of the steel and the hardness measured after completion of the cooling of the steel from the austenitizing temperature. These findings are the results of dilatometry and hardness studies on steels with known (measured) chemical compositions published in the literature. These are not data from a repository. The dataset used for modeling is the result of a considerable effort involving the acquisition and digitization of CCT diagrams. We believe that the methodology used to represent the modeling approach and the data used to estimate the model parameters conform to accepted standards in the field.

The modeling results are compared with published research results. In this sense, our work represents a typical data science study. Although the hardness model is important, it is not the focus of this publication. The hardness modeling method, which uses artificial neural networks and hardness research results, has already been published, peer-reviewed, and widely cited. This publication supplements the data and introduces new model parameters, but the modeling methodology remains unchanged, as described in Section 2.

The presentation of the hardness model in this paper is intended to facilitate the reader's analysis of the entire topic. Our aim was to summarize the complete method for identifying the chemical composition of steel in a single publication, of which the hardness model is a part. Therefore, we have only provided basic statistics to evaluate the accuracy of the hardness model. The primary goal of this publication was to present a method for determining the chemical composition of steel. We regret that this aspect was not considered by the Reviewer.

The information on the statistics of the hardness model — mean absolute error, standard deviation of error and correlation coefficient — can be found in Table 3. In our opinion, these values are at least acceptable and confirm the adequacy of the model.

The information on the error ratio can be found in lines 315 to 322 of the manuscript. In our opinion, this information is sufficient. The term 'error ratio' was only used in Figure 3 to shorten the description of the diagram axis.

We would also like to point out that there are no models in the literature that are directly comparable to our model. We discuss this in Section 2. The only model that is comparable to the hardness model we have developed is Maynier's model. Such a comparison is presented in this paper. The work of other researchers focuses on the Jominy test, which is easier to perform and less costly, but has significant limitations. We discuss this in our paper in the “Hardness Model” section (lines 123-157 of the manuscript).

Reviewer 4 Report

Comments and Suggestions for Authors

In this work, the authors presented an interesting study that presents a hybrid method for calculating the chemical composition of steel with the required hardness after cooling from the austenitizing temperature. Artificial neural networks and genetic algorithms were used to develop the model. Based on 550 diagrams of continuous cooling transformation (CCT) of structural steels available in the literature, a dataset of experimental data was created. Artificial neural networks were used to develop a hardness model describing the relationship between the chemical composition of the steel, the austenitizing temperature, and the hardness of the steel after cooling. A genetic algorithm was used to identify the chemical composition of the steel with the required hardness. The value of the objective function was calculated using the neural network model. The developed method for identifying the chemical composition was implemented in a computer application. Examples of calculations of mass concentrations of steel elements with the required hardness after cooling from the austenitizing temperature are presented.

The paper is interesting. However, some points should be considered to improve the quality of the document:

1. The authors generalize the term "steel" in the study, implying that it is applicable to all existing steels.

2. The hardness model is limited to the range of mass concentrations of the elements shown in Tables 1 and 2. In this sense, the authors should limit the study to only the steels that satisfy the range of elements according to the steel standards selected for this study.

3. In Table 9. The chemical composition of C (wt %) is presented as 0.04 for Solution 2. Also, in Table 11, Ni (wt%) for solution 1 is presented as 0.03. Please clarify these values.

4. The hardness values for solutions from 1 to 5 are consistent in Figures 6, 7 and 8. However, it’s not clear to me why the results of chemical composition for solutions from 1 to 5 of the three examples have a wide range of dispersion of error. I´m concerned about the chemical composition results with the model proposed in this study. Can the authors clarify this variability in results for the three examples for solutions from 1 to 5 in Tables 7, 9 and 11?

5. The authors do not validate their simulated results with experimental verification and only mention that it is future work. Please provide experimental results at least of the three cases analyzed.

Author Response

Dear Reviewer,

Thank you for your comments on our manuscript entitled “A Hybrid Method for Calculating the Chemical Composition of Steel with the Required Hardness After Cooling from the Austenitizing Temperature”. We have carefully studied the comments and prepared explanations which we hope will meet with your approval. We would like to thank the reviewer again for his comments. We appreciate the time and effort you have taken to provide feedback on our manuscript and are grateful for the insightful comments.

Best regards
Authors

Comments and Suggestions for Authors

In this work, the authors presented an interesting study that presents a hybrid method for calculating the chemical composition of steel with the required hardness after cooling from the austenitizing temperature. Artificial neural networks and genetic algorithms were used to develop the model. Based on 550 diagrams of continuous cooling transformation (CCT) of structural steels available in the literature, a dataset of experimental data was created. Artificial neural networks were used to develop a hardness model describing the relationship between the chemical composition of the steel, the austenitizing temperature, and the hardness of the steel after cooling. A genetic algorithm was used to identify the chemical composition of the steel with the required hardness. The value of the objective function was calculated using the neural network model. The developed method for identifying the chemical composition was implemented in a computer application. Examples of calculations of mass concentrations of steel elements with the required hardness after cooling from the austenitizing temperature are presented.

The paper is interesting. However, some points should be considered to improve the quality of the document:

  1. The authors generalize the term "steel" in the study, implying that it is applicable to all existing steels.

Thank you for pointing this out. We have added the name of the steel group in the manuscript: Steels for quenching and tempering (line 112).

  1. The hardness model is limited to the range of mass concentrations of the elements shown in Tables 1 and 2. In this sense, the authors should limit the study to only the steels that satisfy the range of elements according to the steel standards selected for this study.

Thank you for the suggestion, but we have a different opinion on this issue. We do not limit modeling to steels covered by standards. The model can support the design of new steel grades and in our opinion should not be limited to the steel grades described in the standards. We also use published research results of experimental steels that are not standardized in our modeling.

  1. In Table 9. The chemical composition of C (wt %) is presented as 0.04 for Solution 2. Also, in Table 11, Ni (wt%) for solution 1 is presented as 0.03. Please clarify these values.

Thank you for pointing this out. In Table 9, the mass concentration of C (wt%) should be 0.4. In Table 11, the mass concentration of Ni(wt%) should be 0.30. Mistakes have been corrected.

  1. The hardness values for solutions from 1 to 5 are consistent in Figures 6, 7 and 8. However, it’s not clear to me why the results of chemical composition for solutions from 1 to 5 of the three examples have a wide range of dispersion of error. I´m concerned about the chemical composition results with the model proposed in this study. Can the authors clarify this variability in results for the three examples for solutions from 1 to 5 in Tables 7, 9 and 11?

Thank you for this suggestion. In fact, the reader can see that there is a large dispersion of error values. The tables show the sum of the absolute error values for five cooling rates. We have made changes to the article that explain the method used to calculate this error.

  1. The authors do not validate their simulated results with experimental verification and only mention that it is future work. Please provide experimental results at least of the three cases analyzed.

Thank you for pointing this out. The comparison of the calculated mass concentration of a single element (Cr) with experimental data is described in Example 1. We did not carry out our own experiment, but compared the calculation results with information available in the literature. Detailed information can be found in the manuscript. In Example 2, we compared the calculated mass concentrations of two elements (Cr and Mn) with experimental data. Please note that in this case there is a relationship between the concentration of chromium and manganese. The sum of the concentrations of these elements is close to each other and corresponds to the concentration in the compared steel. In order to verify the calculated mass concentrations of all analyzed elements, we need to perform our own research. These research procedures should include the following: Melting of steels with the required chemical composition. The next step should be the plastic processing of these steels. Then carrying out heat treatment with hardening and hardness measurements. These studies are time-consuming and costly. As mentioned in the summary, these studies are planned and will be carried out after developing the model for calculating the chemical composition of steel with the required CCT diagram and obtaining the necessary financial resources.

Round 2

Reviewer 1 Report

Comments and Suggestions for Authors

well revised, can be accepted in present form.

Reviewer 3 Report

Comments and Suggestions for Authors

I propose publishing the article

Reviewer 4 Report

Comments and Suggestions for Authors

The article is interesting and well organized and described. The authors have addressed the comments. It is recommended that the article be published in its present form.